# The Glucose Sensitivity of Insulin Secretion-Lessons from In Vivo and In Vitro Studies in Mice

**DOI:** 10.3390/biom12070976

**Published:** 2022-07-12

**Authors:** Bo Ahrén

**Affiliations:** Department of Clinical Sciences Lund, Lund University, 22185 Lund, Sweden; bo.ahren@med.lu.se

**Keywords:** glucose, insulin secretion, in vivo, isolated islets, mice

## Abstract

This study explored the relationship between the glucose dose and insulin response from beta cells in vivo and in vitro in mice. Glucose was administered intravenously at different dose levels (from 0 to 0.75 g/kg) in anesthetized C57BL/6J mice, and the glucose and insulin concentrations were determined in samples taken after 50 min. Furthermore, freshly isolated mouse islets were incubated for 60 min in the presence of different concentrations of glucose (from 2.8 to 22.2 mmol/L) and insulin levels were analyzed in the medium. It was found that insulin levels increased after an intravenous injection of glucose with the maximal increase seen after 0.35 g/kg with no further increase after 0.5 or 0.75 g/kg. The acute increase in insulin levels (during the first 5 min) and the maximum glucose level (achieved after 1 min) showed a curvilinear relation with the half-maximal increase in insulin levels achieved at 11.4 mmol/L glucose and the maximal increase in insulin levels at 22.0 mmol/L glucose. In vitro, there was also a curvilinear relation between glucose concentrations and insulin secretion. Half maximal increase in insulin concentrations was achieved at 12.5 mmol/L glucose and the maximal increase in insulin concentrations was achieved at 21.5 mmol/L. Based on these data, we concluded that the glucose-insulin relation was curvilinear both in vivo and in vitro in mice with similar characteristics in relation to which glucose levels that achieve half-maximal and maximal increases in insulin secretion. Besides the new knowledge of knowing these relations, the results have consequences on how to design studies on insulin secretion to obtain the most information.

## 1. Introduction

Defective glucose-stimulated insulin secretion is a main characteristic of type 2 diabetes and a target for glucose-lowering therapy [1,2,3,4]. Therefore, models to explore the islet effects of glucose are important for understanding the normal physiology, pathophysiology and islet-target therapy. Glucose is known to trigger a rapid first phase secretory response from the beta cells, which results in extracellular discharge of a rapid releasable pool of insulin-containing secretory granules [5,6]. The signaling behind this effect involves an initial phosphorylation of the glucose molecule by glucokinase [7]. This, subsequently, results in a chain of metabolic reactions leading to the closure of ATP-dependent potassium channels, the opening of calcium channels, and an inflow of calcium into the cells with a raise of intracellular cytosolic calcium concentrations that triggers the extracellular translocation of the secretory granules [4,5]. Glucose also amplifies an already stimulated insulin secretion, which is an effect that is seen after the first triggering phase has passed [5]. This later second phase relies mainly on cyclic AMP, the level of which is increased by the activation of adenylate cyclase, although also other mechanisms are involved [8,9].

The glucose concentrations to stimulate insulin secretion are in the mmol/L range, which is a range that beta cells normally are exposed to, as evident from 24 h studies in both man [10] and mice [11]. Studies to characterize the glucose effects have been undertaken in vitro after incubation or perifusion of isolated islets in different glucose concentrations. Such studies have demonstrated that there is a sigmoidal relation between glucose concentrations and insulin secretion with the glucose threshold for stimulation of insulin secretion in the range of 3–5 mmol/L and maximal insulin secretion achieved at >20 mmol/L [5,12,13,14,15]. It has also been demonstrated that the relation between glucose concentrations and insulin secretion is right-shifted in mouse islets compared to human islets, resulting in different glucose values for a half-maximal insulin secretory effect between humans (7.9 mmol/L) and mice (13.7 mmol/L) [14]. This species difference has been suggested to be caused by differences in expression of glucose transporters [16]. This species difference is also consistent with higher fasting glucose levels in mice (6–8 mmol/L; [11,17]) than in humans (4–6 mmol/L; [10]). However, whether a similar sigmoidal relation between glucose levels and insulin secretion exists also in vivo is not known, since the studies exploring this relation have never compared the in vitro versus in vivo condition. In fact, the relation might be different in vivo, since the beta cells are exposed to several factors in vivo, such as circulating nutrients and hormones, nerves and paracrine islet effects, which may perturb the beta cell response to glucose.

Therefore, this study was undertaken in order to explore the relationship between the glucose dose (level) and insulin response (secretion) from beta cells by performing standardized model experiments using several different glucose doses in mice both in vivo and in vitro. For that reason, a series of experiments were performed in mice where glucose was administered intravenously at doses ranging from 0 to 0.75 g/kg and insulin levels were measured and related to the glucose concentrations. Furthermore, a series of in vitro studies on incubated freshly isolated islets was performed with varying glucose concentrations in the medium from 2.8 to 22.2 mmol/L.

## 2. Materials and Methods

### 2.1. Mice

All experiments were undertaken in female C57BL/6J mice (Taconic, Skensved, Denmark; 4–6 months of age). The animals were maintained in a temperature-controlled room (22 °C) on a 12:12 h light-dark cycle (light on at 7:00 AM). Mice were fed a standard pellet diet (total energy 14.1 MJ/kg with 14% from fat, 60% from carbohydrate and 26% from protein; SAFE, Augy, France) and tap water ad libitum. During experimental days, food was removed from the cages at 7:30 AM and the actual experiments started at 12:30, i.e., during the light cycle. We used female mice only to avoid the stress of single housing, which is used in male mice, and to be in line with the previous study on insulin response to oral glucose at different doses [18]. We used the mice randomly during the estrous cycle.

### 2.2. Animal Disposition

A total of 67 animals were allocated for the in vivo studies and 18 animals for the in vitro studies. In vivo, studies were undertaken in batches of 6–8 mice on each experimental day by one experienced technician. All individual results from the completer population were included in the final analysis and statistics.

### 2.3. Experiments

**In vivo:** After a 5-h fast, mice were anesthetized with a fixed dose combination of fentanyl (0.02 mg/mouse)-fluanisone (0.5 mg/mouse) and midazolam [19]. After 15 min, mice were given glucose (0.125 to 0.75 g/kg, dissolved in saline) or saline alone (i.e., 0 glucose) intravenously as a 3 sec injection in a tail vein (injection volume 10 µL/g body weight). Whole blood was sampled in heparinized pipettes from the intraorbital retrobulbar sinus plexus (40 µL) at 0, 1, 5, 20, and 50 min. Plasma was separated by centrifugation and stored at −20 °C until analysis for insulin.

**In vitro:** Pancreatic islets were isolated by collagenase digestion and handpicked under a microscope. Batches of freshly isolated islets were first preincubated for 60 min in a HEPES (4-(2-hydroxyethyl)-1-piperazineethanesulfonic acid) balanced salt solution containing 125 mmol/L NaCl, 5.9 mmol/L KCl, 1.28 mmol/L CaCl_2_, 1.2 mmol/L MgCl_2_, 25 mmol/L HEPES (pH 7.4), 5.6 mmol/L glucose, and 0.1% fatty acid free BSA (Boehringer Mannheim, Mannheim, Germany) at 37 °C for 60 min. Thereafter, islets in groups of three were incubated in 200 µL of the above described buffer with different glucose concentrations, ranging from 2.8 to 22.2 mmol/L at 37 °C during the 60 min. Aliquots of the buffer were collected and stored at −20 °C until analysis for insulin.

### 2.4. Assays

Glucose was detected with the glucose oxidase method using Accu Chek Aviva (Hoffman-La Roche, Basel, Switzerland). Insulin was determined by ELISA (Mercodia, Uppsala, Sweden). The intra-assay coefficient of variation (CV) of the method was 4% at both low and high levels and the interassay CV was 5% at both low and high levels. The lower limit of quantification of the assay was 6 pmol/L.

### 2.5. Data and Statistical Analysis

Data were reported as means ± SEM. Areas under the glucose (AUC_glucose_) and insulin (AUC_insulin_) curves were calculated using the trapezoid rule for the entire 50 min study period. AIR (acute insulin response) was calculated as the suprabasal mean of 1 and 5 min insulin levels (i.e., mean of 1 and 5 min insulin levels minus baseline insulin levels). The glucose elimination rate (K_G_) was calculated as the percentage reduction in glucose levels per minute between 1 and 20 min following the glucose injection after the logarithmic transformation of the glucose data. These time points were chosen because glucose levels peaked at 1 min in all mice were given glucose and, therefore, glucose elimination would be standardized from the peak value. Furthermore, the mean of 1 and 5 min was chosen for AIR because although the peak insulin after glucose was seen at 1 min in all mice given glucose, the 5 min values were often above baseline, and, therefore, the acute insulin release occurred during the first 5 min. The curve fitting was carried out using Sigmaplot, v. 14.5. For all analyses, statistical significance was defined as *p* < 0.05.

## 3. Results

### 3.1. Intravenous Glucose

Figure 1 shows the glucose and insulin responses to intravenous glucose injection at different doses. Table 1 shows the estimated values for AIR, AUC_insulin_, AUC_glucose_, and K_G_. It was seen that glucose levels increased dose-dependently after the glucose injection, with a peak at 1 min, followed by a gradual return to baseline. Glucose elimination (K_G_) was higher by increasing the glucose dose. Further, insulin levels increased promptly with a peak at 1 min after glucose injection. Figure 1 also shows that the relation between the increase in AIR and the 1 min glucose level was curvilinear. The curve fitting, which was normalized for the change in insulin levels after saline injection, showed a quadratic relation between the measures as the best fit, with the equation y = 24.4x − 0.55 ×^2^ − 154 (r = 0.9573), where y was AIR in percentage of maximal AIR (after 0.75 g/kg) and x was the 1 min glucose level. Based on this curve, the half-maximal increase in AIR was seen at 11.4 mmol/L and the maximal AIR was seen at 22.0 mmol/L.

### 3.2. Incubation of Islets

Figure 2 shows the insulin levels after incubation of isolated islets in different concentrations of glucose. It was seen that there was a dose-dependent increase in insulin. The relation between insulin and glucose was curvilinear with a cubic relation as the best fit. The equation of this curve was y = 25.6 − 10.1x + 1.4 ×^2^ − 0.037 ×^3^ (r = 0.9982), where y was the insulin concentration (pmol/L) and x was the glucose concentration (mmol/L). This means that y would never reach 0, that half maximal increase in insulin concentrations was achieved at 12.5 mmol/L, and that the maximal insulin concentrations was achieved at 21.5 mmol/L.

## 4. Discussion

This study explored the glucose sensitivity in beta cells under experimental conditions in model experiments in vivo and in vitro in mice. The relation between glucose dose and insulin response was similarly curvilinear in vivo and in vitro with a half-maximal effect to stimulate insulin secretion achieved at 11.4 mmol/L in vivo and at 12.5 mmol/L in vitro. Furthermore, similar glucose levels induce maximal insulin secretion in vivo (22.0 mmol/L) and in vitro (21.5 mmol/L). Previous studies have similarly demonstrated a curvilinear relation between glucose concentrations and insulin secretion in isolated islets from mice [5,12,13,14,16,20] and humans [14,16,21]. Our results, therefore, suggested that the beta cell sensitivity to glucose was not different in vivo versus in vitro, and we confirmed that glucose levels eliciting half maximal effect in mice was higher than previously reported in humans [14,16]. It should be emphasized that the relation between glucose and insulin response was estimated during the initial 5 min after glucose administration and thus relevant for the first phase of insulin secretion. Whether a similar relation existed for the second phase of insulin secretion remained to be established. It was also possible to calculate a relation between AUC_glucose_ and AUC_insulin_, which showed a half maximal effect on AUC_insulin_ by 104 mmol/L min in AUC_glucose_, which was achieved after ≈0.25g/kg glucose. However, this relation included not only the effect of glucose to stimulate insulin secretion but also the effects of insulin to reduce glucose levels and was, therefore, less relevant for the conclusion on glucose sensitivity of beta cell secretion. It should also be emphasized that the in vivo and in vitro conditions differed, since islets in vivo were regulated by circulating hormones and nutrients, nerves, and paracrine factors, whereas islets in vitro were incubated in a medium with different composition than the in vivo situation.

We have previously published data on the insulin response to different concentrations of glucose when administered through oral gavage [18]. The purpose of that study was to analyze whether the insulin response to oral glucose was altered in mice with genetic deletion of incretin hormone receptors. However, the data in that study could also be compared to our present data to explore whether there was a difference in beta cell glucose sensitivity after oral versus intravenous glucose administration. It should be emphasized, however, that there are fundamental differences between these two experimental conditions. Thus, after oral glucose, there was an increase in levels of incretin hormones GIP and GLP-1 [22,23] which was not seen after intravenous glucose, and furthermore, neural activity may have been initiated [24]. Additionally, there was a marked difference in the rapidity of the rise of glucose levels after intravenous versus oral glucose. To explore this, we revisited these data from the previous study [18] and reanalyzed them for estimation of the best curve fit between the rise of insulin at 15 min after oral glucose (the early insulin response, EIR) and the 15 min glucose level. Figure 3 shows that the relation between glucose and insulin after oral glucose was strikingly different from our present findings after intravenous glucose. After oral glucose, the relation is exponential (y = 9.6 − 2.17x + 0.45 ×^2^; r = 0.9997), where the positive value in front of the squared x indicated that the curve did not reach a maximum. Thus, it was impossible to calculate the glucose concentration eliciting half-maximal insulin secretion. The continuously increased insulin response by raising glucose levels after oral glucose was most likely dependent on the strong effects of the incretin hormones released after oral glucose, GIP, and GLP-1 [22,23], which both markedly augmented glucose-stimulated insulin secretion in mice [25].

The extent by which the rapidity of the increase in glucose levels influenced the insulin response may be important. Thus, the marked increase in 1 min insulin levels after intravenous glucose may be dependent not only on the achieved glucose levels per se, but also on the fact that glucose levels increased promptly and rapidly. We illustrate this in Figure 4 by comparing three different studies when glucose levels had been increased to approximately the same level but with different time patterns. The first study was the present study where glucose was rapidly injected intravenously within 3 s—a very rapid and huge increase in insulin levels was seen. The second study was a hyperglycemic clamp, where glucose levels were gradually increased during 2 min, whereafter they stayed constant [26]. The third study was a hyperglycemic ramp when glucose was slowly increased over 40 min [27]. The absolute increase in glucose levels was approximately the same in these three studies, but there was a difference in time pattern. The resulting increase in insulin levels was lower with a slower increase in glucose levels, particularly notable during the glucose ramp when beta cells responded very slowly and no clear first phase but only a gradual second phase was evident [27]. Nevertheless, under these conditions, it was shown that there was a marked augmentation of glucose-stimulated insulin secretion by GLP-1 [27], which showed that under in vivo conditions, glucose was only one regulator of beta cell function.

Glucose elimination (K_G_) was augmented when the glucose dose was increased (Figure 1). This may be explained by the higher insulin levels seen after higher glucose doses. However, other factors may have contributed as well, considering that the increase in insulin levels after the three highest doses of glucose were similar, despite different glucose levels, and yet, glucose elimination was increased by raising the glucose dose. Such a factor may be glucose itself, since glucose stimulates a process called glucose-dependent glucose elimination, also termed glucose effectiveness [28]. In fact, when completely abolishing any insulin response to intravenous glucose by the drug diazoxide [19,29], there was a clear glucose elimination, which was approximately 50% of that seen when insulin levels were allowed to increase. Another potential contributing factor was that glucose levels above ≈22 mmol/L were higher than the kidney threshold for glucose absorption [30]. Therefore, when glucose levels exceeded this threshold, there was a loss of glucose in the urine, which augmented glucose elimination.

Our present results have implications for the design of studies on glucose-stimulated insulin secretion to capture not only basal and maximal insulin secretion but also the half-maximal insulin secretion. Many studies, particularly in vitro studies, explore the effects of a low and a high glucose dose only, optimally 3–5 mmol/L and 14–15 mmol/L [14]. However, the use of only two doses may overlook the half-maximal effect, which may be of relevance. Figure 5 illustrates this; the normal conditions are curve A. If a study now includes only a low and a high glucose level, the augmented maximal insulin secretion in curve B will be captured, but the reduced beta cell glucose sensitivity in curve C will be overlooked. Additionally, the lower maximal insulin secretion in curve D will be captured by using only a low and high glucose concentration, but the reduced glucose sensitivity will not be found. Therefore, at least three glucose doses are recommended, and our present results will help to select them for studies in mice.

In conclusion, this study in model experiments in mice characterized the relation between the glucose stimuli and the insulin response both in vivo and in vitro. The results showed that the glucose-insulin relation was curvilinear both in vivo and in vitro, with similar characteristics that half-maximal insulin secretion was achieved by approximately 12 mmol/L glucose and maximal insulin response by approximately 22 mmol/L glucose. Besides the new knowledge of knowing these relations, the results have consequences on how to design studies on insulin secretion to obtain the most information.

## Figures and Tables

**Figure 1 biomolecules-12-00976-f001:**
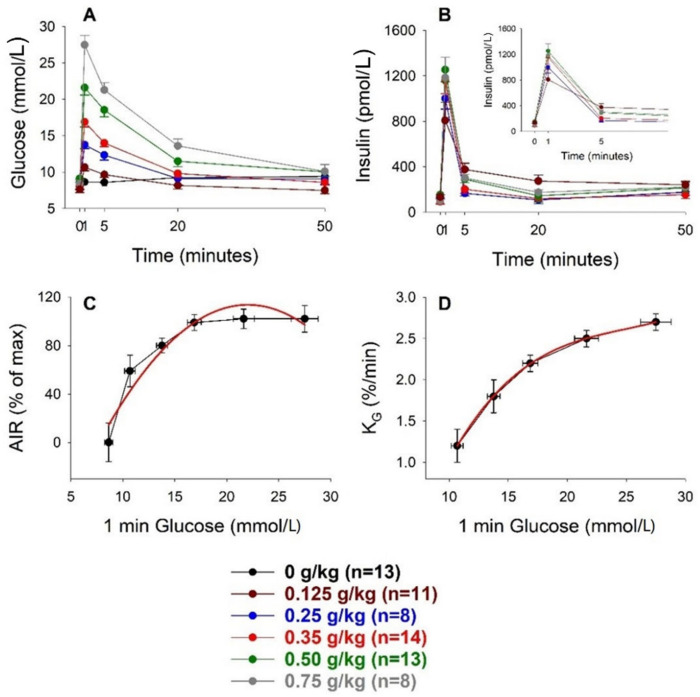
Glucose (**A**) and insulin ((**B**) with insert for initial minutes) levels before and after intravenous administration of glucose (0–0.75 g/kg) and the relation between the 1 min glucose value and acute insulin response (AIR) (**C**) and glucose elimination rate (K_G_) (**D**), respectively, in C57BL/6J mice. Means ± SEM are shown. n indicates the number of animals in each group. Red lines in the two lower panels indicate the best curve fit for the relations.

**Figure 2 biomolecules-12-00976-f002:**
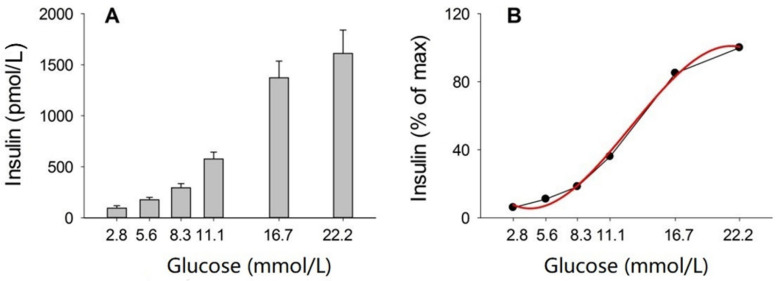
Insulin secretion from islets freshly isolated from C57BL/6J mice stimulated with the indicated glucose concentration in static incubations for 60 min (**A**) and the relation with the best fit shown as the red line (**B**). Means ± SEM are shown in A. N = 8 in each glucose concentration, where n indicates number of studies during which eight incubations were undertaken with three islets in each.

**Figure 3 biomolecules-12-00976-f003:**
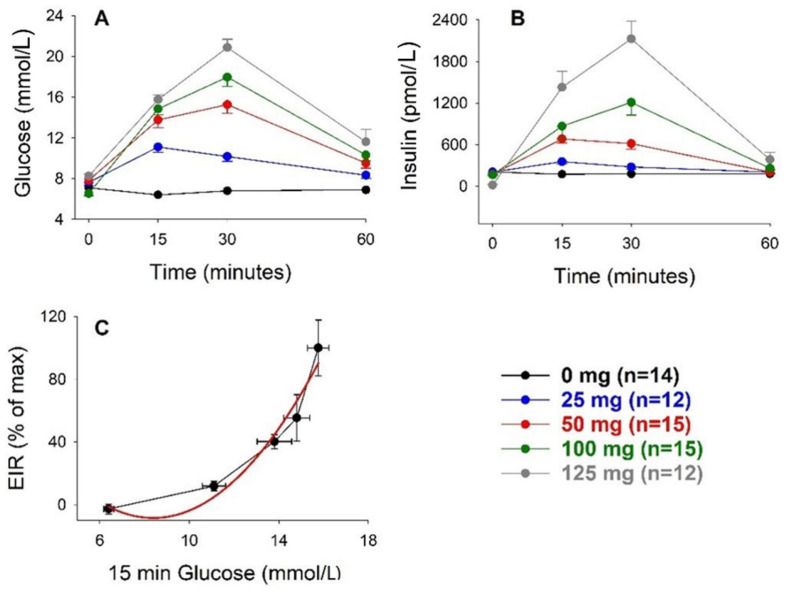
Glucose (**A**) and insulin (**B**) levels before and after oral administration of glucose (0–125 mg per mouse) and the relation between the early insulin response (EIR) and 15 min glucose level in C57BL/6J mice (**C**). Original data reported in [18]. Means ± SEM are shown. n indicates the number of animals in each group. Red line indicates the best curve fit for the relations.

**Figure 4 biomolecules-12-00976-f004:**
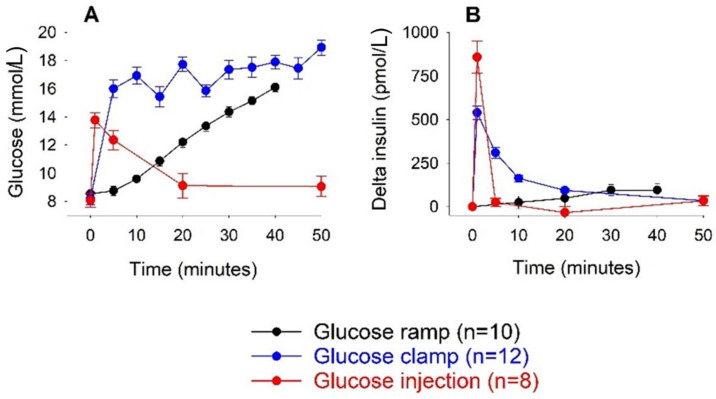
Glucose levels (**A**) and increase in insulin levels (delta insulin) (**B**) during hyperglycaemic clamp (data from [26]), glucose ramp (data from [27]), and intravenous glucose (at 0.35 g/kg) (data from present study) in C57BL/6J mice. Means ± SEM are shown.

**Figure 5 biomolecules-12-00976-f005:**
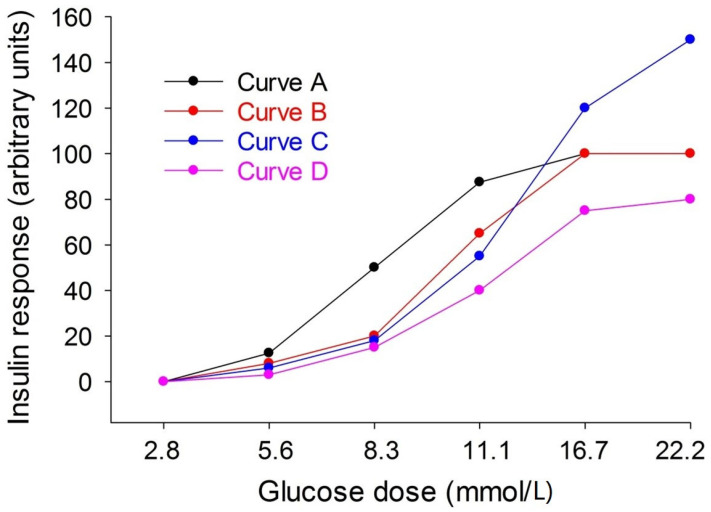
Four different patterns of relation between glucose dose and insulin secretion. For explanation see discussion section.

**Table 1 biomolecules-12-00976-t001:** Body weight, acute insulin response (AIR), suprabasal (incremental) areas under the 0–60 min curves (AUC) for glucose and insulin levels, and glucose elimination rate (K_G_) after intravenous administration of glucose at different dose levels in C57BL/6J mice. Means ± SEM are shown. n indicates the number of animals in each group. NR not relevant.

Glucose Dose (g/kg)	n	Body Weight (g)	AIR (pmol/L)	AUC_glucose_ (mmol/L min)	AUC_insulin_ (nmol/L min)	K_G_ (%/min)
0	13	21.7 ± 0.5	−62 ± 8	18 ± 43	−0.5 ± 1.1	NR
0.125	11	21.9 ± 0.6	326 ± 45	41 ± 16	4.3 ± 1.6	1.2 ± 0.2
0.25	8	21.7 ± 0.8	441 ± 50	99 ± 26	5.5 ± 1.8	1.8 ± 0.2
0.35	14	22.0 ± 0.8	588 ± 53	141 ± 14	8.4 ± 1.1	2.2 ± 0.1
0.50	13	22.1 ± 0.7	619 ± 64	200 ± 21	8.4 ± 2.2	2.5 ± 0.1
0.75	8	20.4 ± 0.2	616 ± 92	336 ± 29	11.0 ± 1.5	2.7 ± 0.1

## Data Availability

Original data are available upon reasonable request to the author.

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
