# Peer review of "The Glucose Sensitivity of Insulin Secretion-Lessons from In Vivo and In Vitro Studies in Mice"

_biomolecules, 2022, doi:10.3390/biom12070976_

Round 1

Reviewer 1 Report

This is a well designed and nicely written study which compares for the first time the glucose sensitivity of the beta-cells of the pancreatic islets in vivo and in vitro.

Although many laboratories are experienced with rodent islet work, very few have the experience of the author in studying fast temporal insulin secretory responses using mice in vivo. This is a strength of the paper.

The results are remarkable in showing great similarity between the glucose sensitivity of the insulin secretory machinery in vivo and in vitro, providing justification of the validity of using isolated islets for research.

The author also makes a deep analysis of important factors which are often overlooked in the design and interpretation of many in vivo and in vitro islet studies. The different glucose-insulin relationship between glucose administered iv and orally was particularly striking and emphasises the importance of the entero-insular axis.

I have no major criticisms of this paper but make a few small suggestions.

1. Please specify iv glucose in the abstract

2. Comment on the use of anaesthesia as this may affect beta cell function. How long were the mice anaesthetised? Of course, the similarity between in vivo and in vitro responses suggest that such an effect is not large.

3. Please comment in M&M why choose and compare 5 min for glucose with 1 min for insulin.

4. Would the author like to say something about first and second phase insulin secretion?

5. Line 197 should read 'marked'

Author Response

Thank you for the positive comments and the valuable comments. I have listed my responses to each of the individual comments here:

  1. Please specify ivglucose in the abstract

Response: Done

  1. Comment on the use of anaesthesia as this may affect beta cell function. How long were the mice anaesthetised? Of course, the similarity between in vivo and in vitro responses suggest that such an effect is not large.

Response: Anesthesia was given 15 min before start of experiments, and the experiments lasted for 50 minutes. This is now stated in the M&M section.

  1. Please comment in M&M why choose and compare 5 min for glucose with 1 min for insulin.

Response: These time points were chosen because glucose levels peaked at 1 min in all mice given glucose and therefore glucose elimination is standardized from the peak value. Furthermore, the mean of 1 and 5 min was chosen for AIR because although the peak insulin is seen at 1 min in all mice given glucose, also the 5 min values are often above baseline, and therefore the acute insulin release occurs during the first 5 minutes, This is now explained in the M&M section.

  1. Would the author like to say something about first and second phase insulin secretion?

Response: First and second phases of insulin secretion are now mentioned in the introduction.

  1. Line 197 should read 'marked'

Response: has been changed

Reviewer 2 Report

Ahren studies the pharmacological disposition of insulin and glucose using both in vivo and in vitro systems.  They compare these results with their previous studies for interpretative purposes. Overall, the manuscript was easy to follow and provides some new information that adds to the field.  I have comments below for revision.

Major suggestions:

1) Suggest measure of plasma GIP and GLP-1 in the current study.

2) Suggest adding another treatment in mice where a second bolus of glucose is provided, i.v. to see if insulin levels are able to increase further to a level similar to what is observed with oral treatment, e.g., about 2000 pM (p.o.) versus about 1200 pM (iv).  

Minor findings for correction:

1. Please include the volume of glucose injected into the mice (lines 92-95).

2. Line 25 "the" missing the "t"

3. Line 61 "has" should be "have"

4. Line 63 "being" should be replaced with "such as"

5. Line 90 "completer" should be "complete"

6. Line 100- "HEPES-balanced salt solution"?

7. Line 110 delete extraneous paragraph return. 

8. Line 116, the description for AIR is confusing.  Please reword. 

9. Figure 1- Change X and or Y axes on 1B so that it is easier to see separation between groups. 

10. When discussing oral treatments from a previous study, please indicate if the glucose was provided by gavage or another means. 

11. Line 178- "raise" should be replaced with the word, "the rise". 

Author Response

Thank you for these positive general comments and the individual comments. I have listed my responses to each of them here..,

Major suggestions:

  • Suggest measure of plasma GIP and GLP-1 in the current study.

Response: Thank you for this suggestion. We have not, however, measured GIP and GLP-1 levels since the study aimed at examining the insulin response to intravenous glucose, after which levels of the incretin hormones are not altered.

2) Suggest adding another treatment in mice where a second bolus of glucose is provided, i.v. to see if insulin levels are able to increase further to a level similar to what is observed with oral treatment, e.g., about 2000 pM (p.o.) versus about 1200 pM (iv).  

Response: hank you for this interesting suggestion. It would indeed be a good follow-up study to examine the absolute relevance of incretin hormones in relation to glucose. However, to inject a second bolus of glucose is probably not the best design, since glucose levels after oral glucose is more sustainely elevated. Therefore, matching glucose levels through a variable glucose administration would probably be the best approach. This is, however, beyond the scope of the current study.

  1. Please include the volume of glucose injected into the mice (lines 92-95).

Response: Injection volume is 10 µl/g body weight; now stated in the M&M section.

  1. Line 25 "the" missing the "t"

Response: Has been corrected.

  1. Line 61 "has" should be "have"

Response: Has been corrected

  1. Line 63 "being" should be replaced with "such as"

Response: Has been corrected

  1. Line 90 "completer" should be "complete"

Response: Sorry, but the correct term is completer population

  1. Line 100- "HEPES-balanced salt solution"?

Response: HEPES = (4-(2-hydroxyethyl)-1-piperazineethanesulfonic acid). This is now explained.

  1. Line 110 delete extraneous paragraph return. 

Response: done

  1. Line 116, the description for AIR is confusing.  Please reword. 

Response: done

  1. Figure 1- Change X and or Y axes on 1B so that it is easier to see separation between groups. 

Response: Thank you for this suggestion. In the revised manuscript, an insert in Fig. 1B has been introduced with insulin levels during the initial minutes only, which makes it easier to distinguish between the different curves,

  1. When discussing oral treatments from a previous study, please indicate if the glucose was provided by gavage or another means. 

Response: done

  1. Line 178- "raise" should be replaced with the word, "the rise". 

Response: done

Reviewer 3 Report

In this paper by Ahren the glucose-dependence of insulin secretion in vitro is compared with the one in vivo of C57BL/6J mice. Specifically, the insulin secretion in response to an intravenous injection is compared with the one of statically incubated isolated islets. The author concludes that under both experimental conditions the half-maximal stimulatory concentration is about 12.5 mM and the maximally effective concentration is about 22 mM. It is recommended that in future studies on insulin secretion, not only basal and maximally effective glucose concentrations, but also a concentration in between are used.    

This is a carefully performed investigation with interesting results.

 I have only minor comments.

The stimulation of isolated islets by static incubation is practically constant and is more similar to a hyperglycemic clamp than to a bolus injection. Since the glucose dependence of the AIR and the one of the AUC insulin are apparently different (table 1) I would suggest to calculate the half maximal concentration of this parameter, too, and to include it in the discussion section.

The in vivo data in the graphical abstract and in Fig. 1C show negative values for “Insulin secretion” or for “AIR”, respectively. Since these values result from the control injections (no glucose) I suggest to normalize these values as zero percent of maximal.

There is no mention of a culture period of the islets. Since most studies use cultured islets the islets should be specifically characterized as freshly isolated.

There are a number of typographical errors.

Author Response

Thank you for this positive general comments and the individual suggestions. I have listed my responses to these suggestions here.

 I have only minor comments.

The stimulation of isolated islets by static incubation is practically constant and is more similar to a hyperglycemic clamp than to a bolus injection. Since the glucose dependence of the AIR and the one of the AUC insulin are apparently different (table 1) I would suggest to calculate the half maximal concentration of this parameter, too, and to include it in the discussion section.

Response: Thank you for this suggestion. We have made such a calculation. It showed that half maximal effect on AUCinsulin was achieved by 104 mmol/l min in AUCglucose, which is equivalent to the AUCglucose after ≈0.25g/kg glucose. However, this relation includes not only the effect of glucose to stimulate inulin secretion but also effects of insulin to reduce glucose levels and is therefore less relevant for the conclusion on glucose sensitivity of beta cell secretion. This is now discussed, as requested (first paragraph in the discussion section).

The in vivo data in the graphical abstract and in Fig. 1C show negative values for “Insulin secretion” or for “AIR”, respectively. Since these values result from the control injections (no glucose) I suggest to normalize these values as zero percent of maximal.

Response: Thank you for this very good suggestion. Fig has been redrawn and new curve fitting have been undertaken.

There is no mention of a culture period of the islets. Since most studies use cultured islets the islets should be specifically characterized as freshly isolated.

Response: There was a 60 min preincubation before the studies. This is now explained in the M&M section. Furthermore, that the islets were freshly isolated have now been stated at several places in the revised manuscript.

There are a number of typographical errors.

Response: We are sorry for this and have gone through the manuscript in detail to avoid this.

Reviewer 4 Report

This article by Ahren compares glucose stimulated insulin secretion in pancreatic islets in vivo and in vitro. The results showed that they exhibited similar secretion capacity at half-maximal insulin secretion and at maximal insulin secretion. These results are of great interest and will be very useful for researchers who will conduct insulin secretion experiments in mice in the future. One concern, however, is that the islet environment in vivo and in vitro is very different. In other words, serum and buffer contain completely different amino acids, lipids, and various proteins such as cytokines, all of which are known to have a significant effect on insulin secretion. Therefore, it is recommended that this point be mentioned as a limitation in the Discussion.

Author Response

Thank you for the positive general comments and the suggestion, which I have responded to here.

This article by Ahren compares glucose stimulated insulin secretion in pancreatic islets in vivo and in vitro. The results showed that they exhibited similar secretion capacity at half-maximal insulin secretion and at maximal insulin secretion. These results are of great interest and will be very useful for researchers who will conduct insulin secretion experiments in mice in the future. One concern, however, is that the islet environment in vivo and in vitro is very different. In other words, serum and buffer contain completely different amino acids, lipids, and various proteins such as cytokines, all of which are known to have a significant effect on insulin secretion. Therefore, it is recommended that this point be mentioned as a limitation in the Discussion.

Response: Thank you for this suggestion. This aspect is now discussed (last sentence of first paragraph of discussion section).

Round 2

Reviewer 2 Report

Revisions are appropriate

Author Response
Thanks.